# Computational thinking in university students: The role of fluid intelligence and visuospatial ability

**Gabor Aranyi**[1,2]*, **Kristof Kovacs**[3], **Ferenc Kemény**[1,4], **Orsolya Pachner**[1], **Balázs Klein**[5], **Eszter P. Remete**[1]

1 Faculty of Education and Psychology, ELTE Eötvös Loránd University, Budapest, Hungary, 2 Faculty of Psychotherapy Science, Sigmund Freud Private University, Vienna, Austria, 3 Institute of Psychology, ELTE Eötvös Loránd University, Budapest, Hungary, 4 Institute of Psychology, University of Graz, Graz, Austria, 5 Testar Ltd., Luton, United Kingdom

* aranyi.gabor@ppk.elte.hu

**Data Availability Statement:** All relevant data are within the manuscript and its Supporting Information files.

## Abstract

Computational thinking (CT) is a set of problem-solving skills with high relevance in education and work contexts. The present paper explores the role of key cognitive factors underlying CT performance in non-programming university students. We collected data from 97 non-programming adults in higher education in a supervised setting. Fluid intelligence, crystallized intelligence, and visuospatial ability were assessed using computerized adaptive tests; CT was measured using the Computational Thinking test. The direct and indirect effects of gender and visuospatial ability through fluid intelligence on CT were tested in a serial multiple mediator model. Fluid intelligence predicted CT when controlling for the effects of gender, age, and visuospatial ability, while crystallized intelligence did not predict CT. Men had a small advantage in CT performance when holding the effects of cognitive abilities constant. Despite its large correlation with gender and CT, visuospatial ability did not directly influence CT performance. Overall, we found that programming-naive computational thinkers draw on their reasoning ability that does not rely on previously acquired knowledge to solve CT problems. Visuospatial ability and CT were spuriously associated. Drawing on the process overlap theory we propose that tests of fluid intelligence and CT sample an overlapping set of underlying visuospatial processes.

## Introduction

### Background: Computational thinking

Computational Thinking (CT) is a group of problem-solving abilities that are especially relevant in a world where computers are ubiquitous and everyday activities are increasingly carried out using computers. The concept of CT originates in computing [1,2]; however, its scope and usefulness extend beyond solving problems using computers. Indeed, many argue that developing CT skills does not even necessarily require the use of computers [3,4]. It is generally accepted in Computer Science (CS) and Computer Science Education (CSE) literature that CT

**Funding:** Gabor Aranyi received funding by the National Research, Development and Innovation Office of Hungary (NKFIH): Grant OTKA-FK-143095 project entitled 'Advancing the Concept of Computational Thinking'. Kristof Kovacs received funding by the National Research, Development and Innovation Office of Hungary: Grant FK-21-138971, by the János Bolyai Research Scholarship of the Hungarian Academy of Sciences and by the ÚNKP-23-5 New National Excellence Program of the Ministry for Innovation and Technology from the source of the National Research, Development and Innovation Fund. Balázs Klein is the executive director of Testar Ltd. The funders provided support in the form of salaries for authors, but had no role in study design, data collection and analysis, decision to publish, or preparation of the manuscript.

**Competing interests:** Balázs Klein is the executive director of Testar Ltd. This does not alter our adherence to all PLOS ONE policies on sharing data and materials. The paper reflects the views of the scientists and not the company. The remaining authors declare that the research was conducted in the absence of any commercial or financial relationships that could be construed as a potential conflict of interest.

plays a fundamental role in adapting to and thriving in a data-driven environment organized and supported by information systems.

Although several countries have organized various working groups to disseminate information on how to promote CT [5] and the academic literature abounds with CT-related learning and teaching practices (for a review, see [6]), a unified definition of CT is lacking, partly due to inconsistencies in measurement and an insufficiently established nomological network [7]. In a systematic review of empirical CT studies, Tang et al. [8] identified two broad approaches to conceptualizing CT in the literature.

One the one hand, some researchers define CT by drawing on computing and programming concepts, which is perhaps most suitable when the focus of research is on programming and education. A notable and widely cited example of this approach is the theoretical framework developed by Brennan and Resnick [9], which conceptualizes CT in the three dimensions of (1) computational concepts (e.g., sequences, loops, and data) that computational thinkers draw on during problem solving; (2) computational practices (e.g., iteration, debugging, and abstraction) they apply; and (3) computational perspectives (e.g., expressing and connecting) relating to the ideas formed about the world and the self during the process. Similarly, with a direct focus on CT in teaching science, technology, engineering, and mathematics (STEM) subjects, Weintrop et al. [10] identified 22 skills classified CT into four categories of practices: data, modelling and simulation, computational problem solving, and systems thinking. Although CT is broadly viewed as a cognitive problem-solving process, researchers taking this approach regard it as a quantifiable learning outcome relating to CT components.

On the other hand, CT is regarded as a competence set composed of knowledge related to the domain of computer programming and problem-solving skills, underpinned by a set of (cognitive) abilities that drive CT performance. This approach is more prominent in research focusing on developing CT as a concept and establishing its nomological network. For example, Shute et al. [11] treat CT as "[. . .] the conceptual foundation required to solve problems effectively and efficiently (i.e., algorithmically, with or without the assistance of computers) with solutions that are reusable in different contexts (p. 142)", and identify six facets of CT: decomposition, abstraction, algorithm design, debugging, iteration, and generalization. Implicit in this conceptualization of CT is the notion that while programming-related knowledge and problem-solving skills should be amenable to education intervention (i.e., one can get better in them with learning and practice), the underlying abilities are traits (i.e., characteristics of a person that are presumed to be stable over time).

This composite approach to defining CT has led to linking it to a wide array of cognitive abilities in various age groups, including numerical/mathematical abilities, language, visuospatial abilities, and general cognitive abilities [12]. While certain CT definitions overlap with similar concepts (e.g., algorithmic thinking; see [13]), other, predominantly operational- and educational-curricular definitions (e.g., Román-González et al. [14]) typically, albeit implicitly, conceptualize CT as a formative construct, the meaning of which is thus influenced by its measurement. Underpinning this approach is the notion supported by a growing body of research and theorizing that CT extends well beyond its underlying abilities: it is not merely 'old wine in a new bottle', but a concept rooted in formerly established constructs influenced by recent and accelerating changes in technology and its impact on society.

While some emphasize the role and relevance of CT in primary and secondary education [12,15], others treat it as a new type of literacy and argue that acquiring adequate skills in effectively and efficiently interacting with information systems is essential to anybody, not just computer programmers [11]. Accordingly, several European countries started incorporating the development of CT into school curricula at various points in K-12 [16], both as a stand-alone subject [17] and as part of STEM education [10], reaching back to pre-school level [18],

while others call for more CT research and assessment in the context of higher education and professional development programs [8]. Although CT is well-studied in K-12, (young) adults in higher education received relatively little attention [19]. Parallel to the educational focus, there is a strong thread of research focusing on CT and its relationships with psychological factors, predominantly cognitive abilities [12,14,20], but also attitudes and perceptions of CT ability [21,22], self-efficacy, and personality [23].

The present research investigates the relationship between CT performance, general intelligence, and visuospatial ability in students in higher education. The following sections provide an overview of CT measurement and the relevant research addressing the role of visuospatial ability and general intelligence in driving CT performance. We argue that a better understanding of CT and its relationship with well-established cognitive factors is supported by controlling for the effects of potential confounds which we identify in previous literature, such as participants' age, their programming experience, and widely-reported gender differences in visuospatial ability and in CT performance.

## CT measurement

CT measurement can be context-dependent and in many cases relies on non-standardized, often single-use methods, such as analysis of digital portfolios [24,25] and project evaluation [26], with only a minority of studies using standardized measurement yielding estimates of reliability and validity, and the majority of research focusing on primary- and middle-school students [8]. However, there has been considerable progress in recent years to develop standardized CT measurement methods suitable for assessing CT performance in different age groups and at various skill levels, such as the Computational Thinking test (CTt; [14]) originally developed for middle-school students, its adaptation to be used with students in high school [27] and primary school [12], the gender-agnostic competent Computational Thinking test (cCTt; [28]) developed for upper primary school (grades 3–6), or the Algorithmic Thinking Test for Adults (ATTA; [29]). These advances in standardized CT measurement support efforts to map the complex relationships between CT and its psychological correlates confounded by age and gender differences.

For example, related to age differences, Tsarava et al. [12] adopted the CTt to the target population of 8–10 years old children with data from 192 third and fourth graders in Germany, and found that complex numerical abilities, verbal reasoning, and nonverbal visuospatial abilities predicted CT performance. Based on their findings and evidence from literature, they argue that numerical reasoning skills are only relevant to CT at the primary school level, verbal abilities at the primary and secondary levels, while the effect of non-verbal reasoning may remain at university level and beyond.

A prominent finding involving standardized CT assessment is the advantage of men over women in CT performance. For example, Román-González et al. [14] found gender differences in CT across five grades to the advantage of boys in in a large sample of secondary school students with small effect size, but they reported gender effects on CT performance without controlling for cognitive abilities measured in the study. It seems plausible that CTt items (as well as other typical CT measures) are biased towards an excessive application of logical and visuospatial tasks. For example, Howland and Good [30] found that girls in 7-8[th] grade produced more complex functions than boys in narrative tasks and reviewed several papers that reported the advantage of girls in various programming tasks. Similarly, Polat et al. [22] also found an advantage of boys in a study of secondary school students across two grades with small effect size and argued that this may be due non-task specific factors, such as differences in academic interest (boys being more interested in technical issues) and creativity [31]. Finke

et al. [20] tested if gender differences in CT can be accounted for by cognitive factors and reported statistically significant gender effects in the same age range while controlling the effects of a broad set of cognitive abilities, such as figural- and numerical reasoning, mental arithmetic, algebraic skills, visualization, mental rotation, and visuospatial short-term memory. It needs to be noted that some studies did not find gender differences; however, these studies often lacked standardized CT assessment [32] or a large enough sample for reliably detecting small effects [33].

Based on the above we argue that to clarify the structural relationships between CT and the theorized underlying abilities, the effects of age, and gender differences (both in CT and the correlated abilities) need to be considered simultaneously. In the following sections, we turn to two such abilities that have been linked to CT and may vary with age and gender: visuospatial ability and general intelligence.

## CT and visuospatial ability

There is strong empirical evidence of CT being associated with visuospatial ability, consistently from early primary school to university level [12]. Román-González et al. [14] observed moderate correlation between CT and performance on the spatial factor of the Primary Mental Abilities (PMA) battery on a large sample of 10-16-year-olds, while Città et al. [34] found positive association between mental-rotation and performance on a coding test in 6-10-year-olds. Tsarava et al. [35] found that performance on visuospatial tasks significantly predicted CT performance in primary school children. Visuospatial abilities have been found to be positively related to CT performance and programming ability in university students [36,37]. To provide a tentative theoretical link between visual abilities and CT, Parkinson and Cutts [38] proposed that visualization and mental rotation are key factors in constructing mental models when understanding programming problems and writing code.

The relative advantage of men in spatial ability in general, and mental rotation tasks in particular, is one of the strongest and most consistent gender-difference findings in the cognitive literature [39–41], which could help in explaining the observed advantage of boys on the CTt [14]. Finke et al. [20] argue that mental rotation, the ability to perform complex rotations in three-dimensional space, is especially relevant to CT performance when assessed with the computational thinking test (CTt; [14]) due to the inherently spatial nature of the test items involving geometric patterns and mazes; however, they found that the simultaneous consideration of multiple visuospatial factors (visualization, figural reasoning, and mental rotation) did not account for gender differences in performance on the CTt.

Since the above studies did consider gender differences in CT (direct effects) but not those in the visuospatial ability when predicting CT (indirect gender effects), it remains unclear if observed gender differences in CT performance are (at least partly) explained by gender differences in visuospatial ability. Furthermore, the level of CT expertise may be confounded with gender; men may be overrepresented in the computing professions [42]. Age, often conceptualized as grade in an education context, also need to be considered, as increasing CT expertise, such as familiarity with computational concepts and practices (see [9]) may lead to a lesser degree of reliance on domain-general abilities in solving CT tasks, while similar to numerical skills and reasoning, the role of visuospatial ability in CT performance may differ between age groups (see the argument by Tsarava et al. [12] above).

## CT and general fluid intelligence

Spearman's notion of a general ability factor (g) accounting for individual differences in different facets of cognitive performance [43] has been shown to be a reliable predictor of education

and job performance, emphasizing the role of general intelligence in the context of high cognitive complexity, information processing, and problem solving [44]. However, the nature of g is subject to ongoing academic debate [45–47].

There are several competing models of the structure of cognitive abilities, but the most accepted contemporary model, and the one that is most influential in test construction, is the Cattell-Horn-Carroll (CHC) model [48]. CHC is a three-stratum model, with several so-called narrow factors (such as quantitative reasoning or lexical knowledge) on the 1st stratum, broad abilities (such as visual processing or processing speed) on the 2nd stratum and g on the 3rd stratum. The most important aspect of CHC is the distinction between broad specific cognitive abilities, which stems from Cattell's distinction of fluid and crystallized intelligence, where the former refers to the ability to solve problems independently from previously accumulated knowledge, while the latter refers to the ability to rely on past knowledge and experience. Under the CHC framework they are referred to as fluid reasoning (Gf) and comprehension/ knowledge (Gc), respectively.

In the Context of CT, Román-González et al. [14] drew on the CHC model [49] and found that fluid reasoning and visual processing predicted computational thinking performance in a sample of 1251 Spanish students from 5th to 10th grade. Finke et al. [20] reproduced their findings in a study of 132 Austrian 7th and 8th grade students, corroborating the role of figural reasoning and spatial abilities (as measured by visualization) in driving CT performance. Both studies found gender differences in CT to the advantage of boys; however, gender differences were not reported in reasoning and visual processing to ascertain the degree to which girls' performance on the CT test was curtailed indirectly as compared to the direct gender effect. Similarly, Tsarava et al. [12] found that complex numerical abilities, verbal reasoning ability, and non-verbal visuospatial abilities predicted CT performance in a sample of 192 German 3rd and 4th graders, and reviewed recent literature indicating links between programming and general intelligence, concluding that further research is needed to understand the transfer effects between (general) cognitive abilities and CT.

While fluid intelligence involves reasoning to solve problems without having to rely on previously acquired knowledge, it is explicit in the conceptualization of CT that it should be sensitive to programming-related knowledge acquired in the process of targeted education (often referred to as computational concepts; [9]). It is indeed a focus of performance tests to aid assessment in education and to inform the development of effective CT education programs [12,27], while other foci include the identification of talent [23] and linking CT to various cognitive psychology constructs [14,20]. Accordingly, (cognitive) abilities and various skills and practices acquired in targeted education are confounded, often tacitly, to various degrees in CT definitions (e.g., [11]), frameworks [9] and measures of performance (e.g., CTt; [14]) or self-assessment (CTS; [21,27]).

Based on the intelligence literature, we propose that the empirically observed association of CT with cognitive abilities can be framed in a process-overlap argument [50], where different tests used to measure CT and its underlying factors sample a range of domain-general executive processes and domain-specific processes in an overlapping manner. CT is thus brought in contact with the 'positive manifold', an all-positive pattern of correlations among performance on diverse cognitive tests, which is perhaps the most replicated finding in all of psychology [51]. Following this argument raises the question of how CT is informed by the already-established range of cognitive factors, at least from a nomological standpoint, educational and developmental aspects notwithstanding. Our current research aims to contribute to this discussion as outlined below.

### Research questions and hypotheses

As discussed above, computational thinking can be broadly defined as a set of problem-solving skills employed in the context of computer programming that relies on a set of cognitive abilities. Apart from its empirically established connections to these cognitive abilities, CT performance is confounded by gender and programming-related knowledge generally targeted in computer-science education (referred to as computational concepts and practices). Although the increase in CT performance across grades is reported both in cross-sectional (e.g., [14,20]) and longitudinal studies both in primary education [52] and at the high-school level [53], the unconditional effect of grade is not routinely considered to control for the unconditional effects of cognitive factors on CT (with notable exceptions; [53]), and the effects of cognitive factors conditioned on skill-related factors, such as grade or programming education, are typically not analyzed (for modelling unconditional and conditional effects, see [54]). Thus, considering the level of skill is expected to inform the association between general abilities and CT, while considering gender effects (notably in visuospatial ability) is expected to shed light on gender effects in CT.

One might expect that acquiring more programming-relevant knowledge and thus programming becoming less and less a novel activity in the context of problem-solving, would lead computational thinkers to rely less on their fluid ability and more on their specific skills and knowledge. Thus, given the mixed nature of CT conceptualizations (ability vs. acquired skill), the degree of programming skill is expected to affect the associations between fluid intelligence and CT: the more skill, the lower the expected correlation between Gf and CT. It is therefore important to consider the level of programming skills when exploring relationships between CT and underlying cognitive abilities.

Level of skill in the current study is controlled by elimination: we only recruited participants with no experience in programming. As most studies related to the associations of cognitive abilities and CT as measured by standardized methods concentrate on children in primary and secondary education, measuring cognitive abilities and CT in the comparatively neglected population of adults allows for reproducing previous findings based on younger participants. Additionally, we treat age as a covariate to control the effects it might exert on the association of the constructs. Based on these, we derived the following research questions (RQ) and corresponding hypotheses (H).

*RQ1: Are there gender differences in computational thinking in adults unfamiliar with programming?*

H1: men perform better in computational thinking then women.

In line with the literature, we also hypothesize an effect of gender on visuospatial ability.

*H2: men perform better in visuospatial ability then women.*

Gender is treated as a control variable in the present study, the effects of which on other study variables are considered when examining the relationship between CT and cognitive factors. Thus, H1 is formulated to test if gender differences in CT often reported in younger participants can be observed in programming-naïve university students, while H2 aims at testing gender differences in visuospatial ability often reported in the cognitive literature. We do not postulate gender differences in fluid intelligence.

*RQ2: How is computational thinking related to visuospatial ability, fluid intelligence, and crystallized intelligence in adults unfamiliar with programming?*

In line with the literature, we hypothesize the following associations between intelligence, visuospatial ability, and CT.

*H3: visuospatial ability is positively related to computational thinking.*

*H4: fluid intelligence is positively related to computational thinking.*

*H5: crystallized intelligence is positively related to computational thinking.*

The testing of the above simple associations is extended by the simultaneous consideration of demographic and cognitive factors in predicting CT performance, according to the following research question.

*RQ3: Do gender differences in visuospatial ability account for gender differences in CT performance?*

*H6: the effect of gender on computational thinking performance is mediated by visuospatial ability.*

While including gender controls its confounding effect on the relationship between CT and cognitive factors, simultaneously testing its direct and indirect effects allows to examine the extent to which it may drive CT performance relative to the cognitive factors. As measures of visuospatial ability, fluid intelligence, and CT all rely on tasks essentially visual and spatial in nature (see the following section), they all sample the same processes to some degree according to the process-overlap argument. We therefore consider if visuospatial ability is a bottleneck of CT performance, albeit with no specific a priori hypothesis.

## Materials and methods

### Instruments

**Computational Thinking test (CTt).** Computational thinking performance was measured with the Computational Thinking test (henceforth: CTt), a standardized psychometric instrument originally developed by Román-González [55] as a performance test for secondary students (10–16 years old) based on the framework of Brennan and Resnick [9] that does not require specific prior knowledge or familiarity with programming languages. CTt is a unidimensional instrument that comprises 28 selected response items (one correct answer out of four options) presented in either a maze (Fig 1) or canvas format that can be completed online in 45 minutes. Román-González et al. [14] validated CTt on a large sample of Spanish

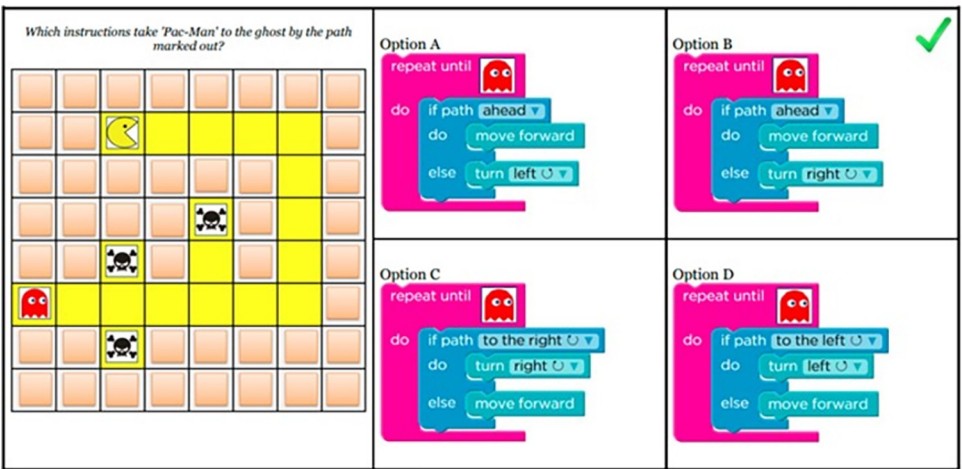

**Fig 1. Sample item of the Computational Thinking Test (CTt).**

secondary students using classical test theory. Since then, the test has been adopted to multiple languages (e.g., Turkish, [22]; German, [27,56]; and Greek, [33]) and its psychometric properties have been tested using both classical test theory and item-response theory [27,57]. We produced a Hungarian version of CTt; initial testing yielded an internal consistency value (Cronbach's α = .79) identical to the one reported by Román-González et al. [14]. To fit the adult population used in the current study, we followed [27] in replacing the five easiest items with five more difficult ones, as advised by the test's author in personal correspondence. Test performance is measured as the sum of correctly solved items out of 28.

**Visuospatial ability (SPOT).** Visuospatial ability was measured with the adaptive visuospatial test SPOT. The test was developed as an adaptive measure of the broad CHC ability Visual processing (Gv). SPOT is a computerized adaptive test (CAT) developed using item-response theory. As opposed to using a set of fixed items, CAT allows for a more precise estimation of participants' abilities and typically reduces testing length by half [58]. At the start of the test, an item of average difficulty is selected; the selection of all subsequent items is dependent on the string of previous responses so that those items are selected from an item pool that yield the most information (high discrimination near the assumed ability level of the test taker). In practice, this means that a more difficult item follows a correct answer, while a mistake results in the presentation of an easier item. The precision of the ability estimate increases with each completed item.

SPOT comprises of 20 items and typically takes 15 minutes to complete. There is a time limit of one minute to each item. Each task features 9 objects in a 3x3 arrangement. A target image is in the middle with red background. The task is to select 3 out of the 8 remaining objects that depict the target object from its possible (rotated) views. The test yields a standardized ability score (theta). A practice item of the SPOT test is presented in Fig 2.

**Scrambled Adaptive Matrices (SAM).** Fluid intelligence (Gf, or fluid reasoning under the CHC model) was measured with the Scrambled Adaptive Matrices (henceforth: SAM). SAM is a computerized adaptive test developed using item-response theory with more than 15000 participants and validated with Raven Progressive Matrices [59]; it is a reliable and valid measure of fluid inductive reasoning.

The testing procedure is identical to that of SPOT described above. Test items consist of 9 elements arranged in a 3x3 matrix. The task is to exchange two of the elements that, if swapped with one another, result in a logical arrangement of elements both horizontally and vertically. The test yields a standardized ability score (theta). Practice items are presented in Fig 3. For information regarding the test's psychometric properties and its validity see [59].

**Non-Directional Vocabulary Test (NoVo).** The third adaptive test measured vocabulary. Vocabulary is a standard measure of Crystallized intelligence [60]–or Comprehension/Knowledge (Gc) under the most recent edition of the CHC model [48]. NoVo stands for Non-directional Vocabulary Test, indicating that the format is different from typical tests of vocabulary in which the examinee has to find the synonym of a target word from a number of options. Instead, NoVo follows the same format as SAM: there are nine words arranged in a 3x3 matrix and the examinee has to indicate which two are closest in meaning. For information regarding the test's psychometric properties and its validity see [61].

## Data collection

Data were collected in 2023, between 1 April and 12 June, at the Institute of Education and Psychology at Szombathely, Faculty of Education and Psychology, Eötvös Loránd University (ELTE PPK), Hungary. The adaptive tests were implemented in an online form by PeopleTest (*www.peopletest.net*); participants received invitation links to the tests, each of which could be

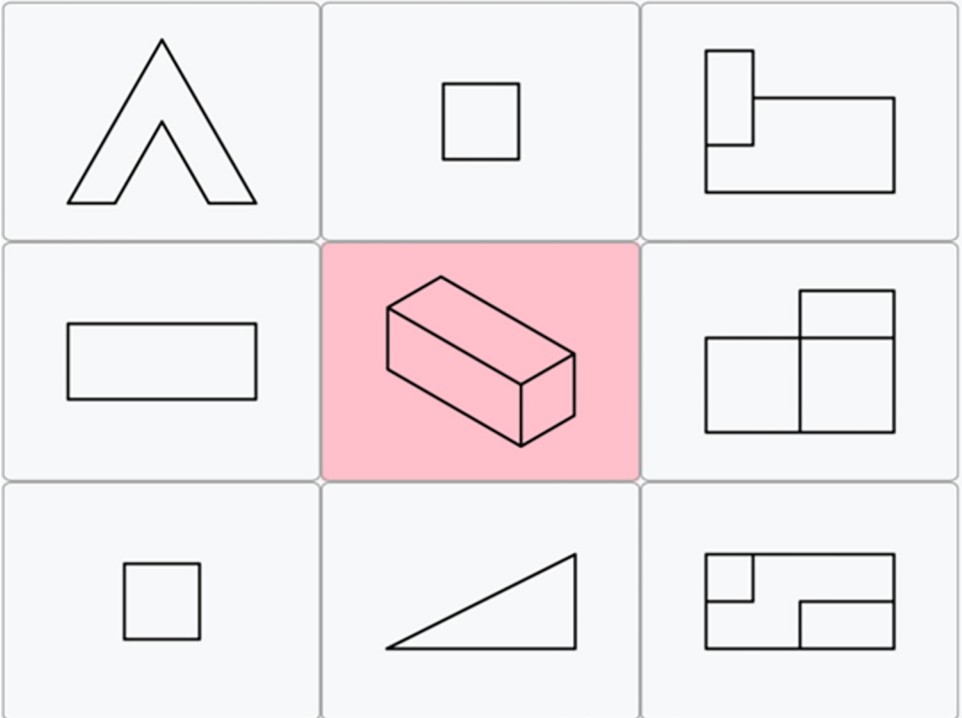

**Fig 2. Practice item of the SPOT test.**

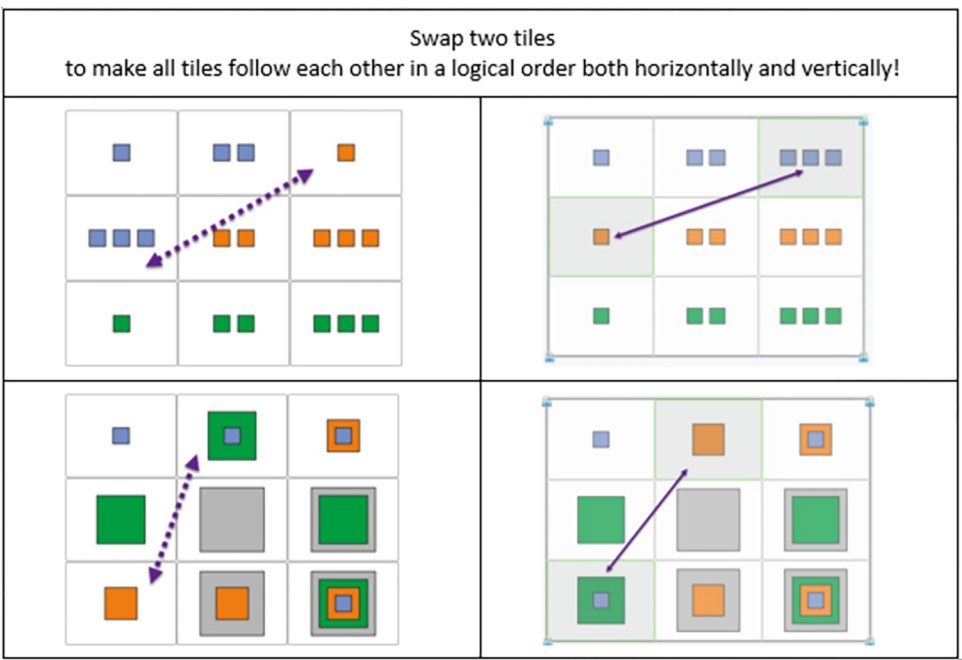

**Fig 3. Practice items of the SAM test.**

attempted once. Demographic information and CTt responses were collected by an online questionnaire developed for the study using the Qualtrics survey tool (*www.qualtrics.com*). A convenience sample was recruited among BA and MA students by research assistants aimed at a minimum of 84 complete responses to support detecting medium effect-sizes ($d$ = .5) with 90% power at the $\alpha$ = .05 significance level. Data were collected individually using a desktop or laptop computer in two sessions lasting a maximum of one hour each to avoid test fatigue. Participants worked individually, under the supervision of a research assistant to ensure no secondary tasks were pursued during testing. During the first session, participants completed the SAM and SPOT tests. The second session consisted of completing the CTt, followed by NoVo and basic demographic questions. Respondents had to be over 18 years old and written consent was collected before participation. All data were collected in the Hungarian language. The study was approved by the ELTE PPK Research Ethics Committee (Reference: 2022/672).

## Participants

Complete data from both sessions was collected from 105 respondents. Eight respondents indicated familiarity with programming; thus, we excluded their data from further analysis, yielding an effective sample size of $N$ = 97; 68 women (70%) and 29 men, which provided 80% power to detect medium effect sizes ($d$ = 0.63) for gender comparisons. Mean age was 28.1 years ($SD$ = 2.6), with a median of 21 and a range between 19 and 31. Eighty-two (85%) were BA students and 15 were enrolled on an MA course.

## Statistical analysis

Statistical analyses were conducted in R (version 4.3.0, [62]), with the packages 'tidyverse' (version 2.0.0, [63]) for data manipulation and visualization, and the 'process' macro (version 4.1.1, [54]) for mediation analysis. We used an alpha level of 0.05 for each statistical test (exact $p$ values are reported); 95% percentile bootstrap confidence intervals are reported for indirect effects, based on 5000 bootstrap samples. For pairwise comparisons, we report Welch's tests with degrees of freedom adjusted for unequal variances [64]. Shapiro-Wilk tests were used for checking normality with a $p < .05$ criterion for assumption violation. We also report Mann-Whitney-Wilcoxon rank-sum test results where the normality assumption was violated. We use Cohen's [65] rules of thumb to interpret effect sizes for the following measures: correlation coefficient ($r$): 0.1 –small, 0.3 –medium, 0.5 –large; Cohen's $d$: 0.2 –small, 0.5 –medium, 0.8 –large.

## Results

### Descriptive analysis and simple associations

The descriptive statistics of Computational Thinking (CTt) performance, and standardized performance scores (theta) achieved on the Visuo-Spatial Ability (SPOT), Scrambled Adaptive Matrices (SAM), and Adaptive Vocabulary (NoVo) tests are presented in Table 1.

 The computerized adaptive tests are based on item response theory (IRT), where the conception of accuracy is different from the concept of reliability in classical test theory. That is, under the IRT framework it is the (item or test) information function that represents precision so that information is the inverse of the items' or the test's accuracy: the more information the test or the items provide, the smaller the error margin of the measurement. Under IRT models the SE (standard error) of each ability (theta) score can be calculated; the higher the information the smaller the SE. We report the mean SEs obtained for each computerized adaptive test in Table 1. Classical reliability coefficients can be approximated from SE values obtained in

**Table 1. Descriptive statistics of CTt and adaptive performance tests (SPOT, SAM, and NoVo).**

| | CTt | | | SPOT | | | SAM | NoVo |
| | Men | Women | TOTAL | Men | Women | TOTAL | TOTAL | TOTAL |
|---|---|---|---|---|---|---|---|---|
| *n* | 29 | 68 | 97 | 29 | 68 | 97 | 97 | 97 |
| Mean | 19.379 | 15.118 | 16.392 | 0.839 | 0.092 | 0.315 | 0.634 | 0.418 |
| *SD* | 4.747 | 3.908 | 4.591 | 0.360 | 0.698 | 0.704 | 0.584 | 0.407 |
| Mean *SE* | | | | | | 0.290 | 0.226 | 0.174 |
| Min | 10 | 8 | 8 | -0.265 | -1.558 | -1.558 | -1.102 | -0.427 |
| Median | 18 | 14.5 | 16 | 0.854 | 0.148 | 0.564 | 0.682 | 0.363 |
| Max | 28 | 23 | 28 | 1.493 | 1.174 | 1.493 | 1.957 | 1.666 |
| *z* skew | 0.242 | 1.653 | 2.037 | -2.689 | -2.021 | -3.371 | -2.355 | 3.061 |
| *z* kurt. | -0.788 | -1.164 | -0.911 | 3.046 | -0.669 | 0.029 | 1.386 | 2.049 |
| *W* | 0.972 | 0.946 | 0.963 | 0.918 | 0.942 | 0.925 | 0.975 | 0.960 |
| *p* (*W*) | 0.615 | 0.005 | 0.008 | 0.027 | 0.003 | 0.001 | 0.063 | 0.005 |

*Notes*. CTt: Computational Thinking test (sum of correctly answered items). SAM: Scrambled Adaptive Matrices (standardized performance score). SPOT: Visuo-Spatial Ability (standardized performance score). NoVo: Adaptive Vocabulary Test. Gender-split descriptives are presented for hypothesised differences only (CTt and SPOT). *W*: Shapiro-Wilk test statistic. *SE*: Standard error of measurement.

IRT [66,67]. The reliability values estimated from the SE values that appear in Table 1 are .92, .95, and .97 for SPOT, SAM, and NoVo, respectively.

Table 2 presents the tests of gender differences on each test, while their correlations are presented in Table 3. The score distributions and gender differences are presented in Fig 4 (CTt) and Fig 5 (SAM and SPOT).

## Hypothesis tests and indirect effects

We tested the structural relationships between computational thinking (CTt), visuospatial ability (SPOT), and fluid intelligence (SAM) in a multiple serial mediation model, where the effect of gender (X) on computational thinking performance (Y) is mediated through visuospatial ability (M1) and fluid intelligence (M2), controlling for the effect of respondents' age (AGE) as covariant (C). Because the simple association between crystallized intelligence (NoVo) and CT was not statistically significant (see Table 3), we did not include NoVo in the mediation model.

This model structure was specified for the following reasons. We selected gender (G) as focal predictor, because as a demographic factor it logically precedes the psychometrically measured, domain-general abilities (visuospatial ability and fluid intelligence) and allows for testing its indirect effect on computational thinking (CTt) as a problem-solving skill through

**Table 2. Tests of gender differences in CTt, SPOT, SAM, and NoVo scores.**

| Hypothesis | Scale | Mean diff. | *t* | *df* | *p* (*t*) | *U* | *p* (*U*) |
|---|---|---|---|---|---|---|---|
| H1 | CTt | 4.262 | 4.259 | 44.956 | < .001 | 481 | < .001 |
| H2 | SPOT | 0.748 | 6.931 | 91.417 | < .001 | 331 | < .001 |
| --- | SAM | 0.260 | 1.961 | 48.693 | .056 | 737 | .050 |
| --- | NoVo | 0.089 | 0.943 | 47.772 | .350 | 886 | .433 |

*Notes*. Test: Welch (degrees of freedom adjusted according to heterogeneity of variances). *U*: Mann-Whitney-Wilcoxon test statistic. Mean differences in SAM, SPOT, and NoVo are men minus women and in standardized performance scores (thetas), thus they can be directly interpreted like Cohen's *d* effect sizes. Cohen's *d* for CTt: 1.03 (large). See Table 1 for descriptive statistics.

**Table 3. Correlations between CTt, SAM, SPOT, NoVo, and gender.**

|  | Gender | SAM | SPOT | CTt |
|---|---|---|---|---|
| SAM | .20*<br>[.01, .39] |  |  |  |
| SPOT | .49**<br>[.32, .63] | .58**<br>[.43, .70] |  |  |
| CTt | .43**<br>[.25, .58] | .56**<br>[.41, .68] | .53**<br>[.37, .66] |  |
| NoVo | .10<br>[-.10, .29] | .12<br>[-.08, .31] | .28**<br>[.09, .46] | .13<br>[-.07, .32] |

*Notes.*

*$p < .05$

**$p < .01$. Gender is coded 0 –women, 1 –men. Fields between gender and other variables are interpreted as point-biserial correlations. Values in braces are limits of 95% confidence intervals (parametric).

visuospatial ability (SPOT) and fluid intelligence (SAM). Linking the mediators in the above order (SPOT to SAM) does not affect statistical inference regarding their influence on computational thinking (CTt); however, it allows for testing if gender differences in fluid intelligence (SAM) we observed in our sample are accounted for by gender differences in visuospatial ability (SPOT). Finally, participants' age was added as covariate to control for its possible (albeit

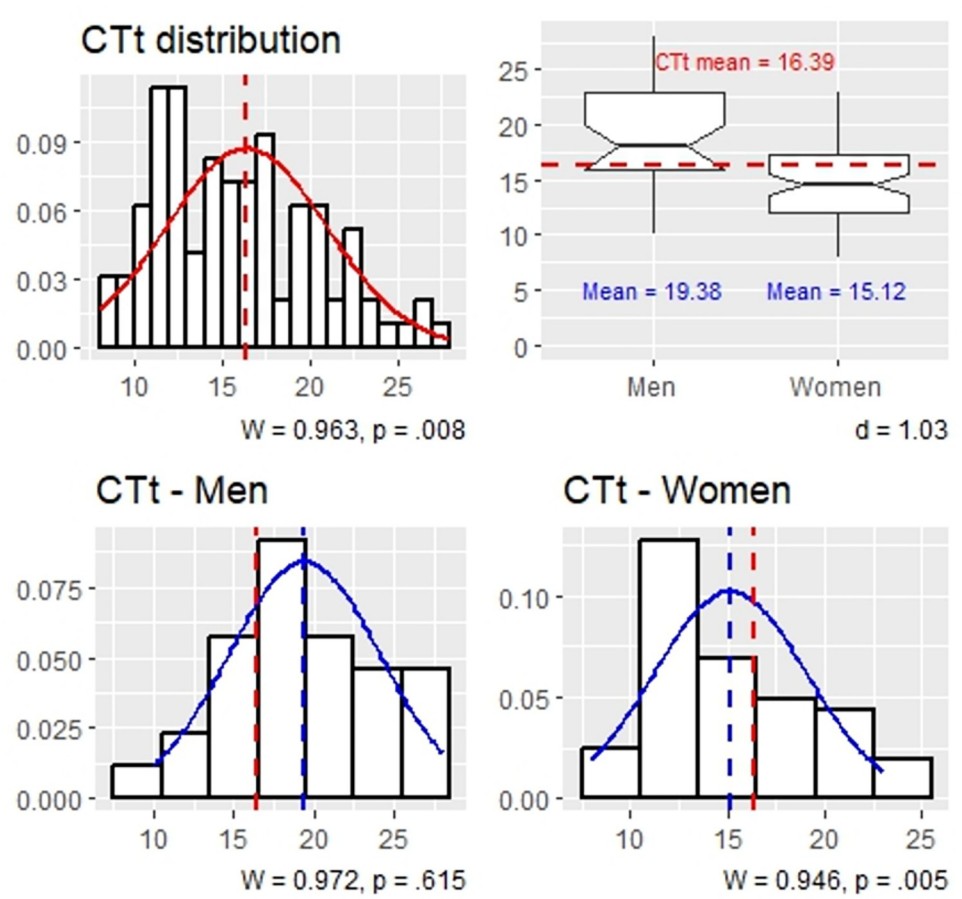

**Fig 4. The distribution of CTt scores in men and women.**

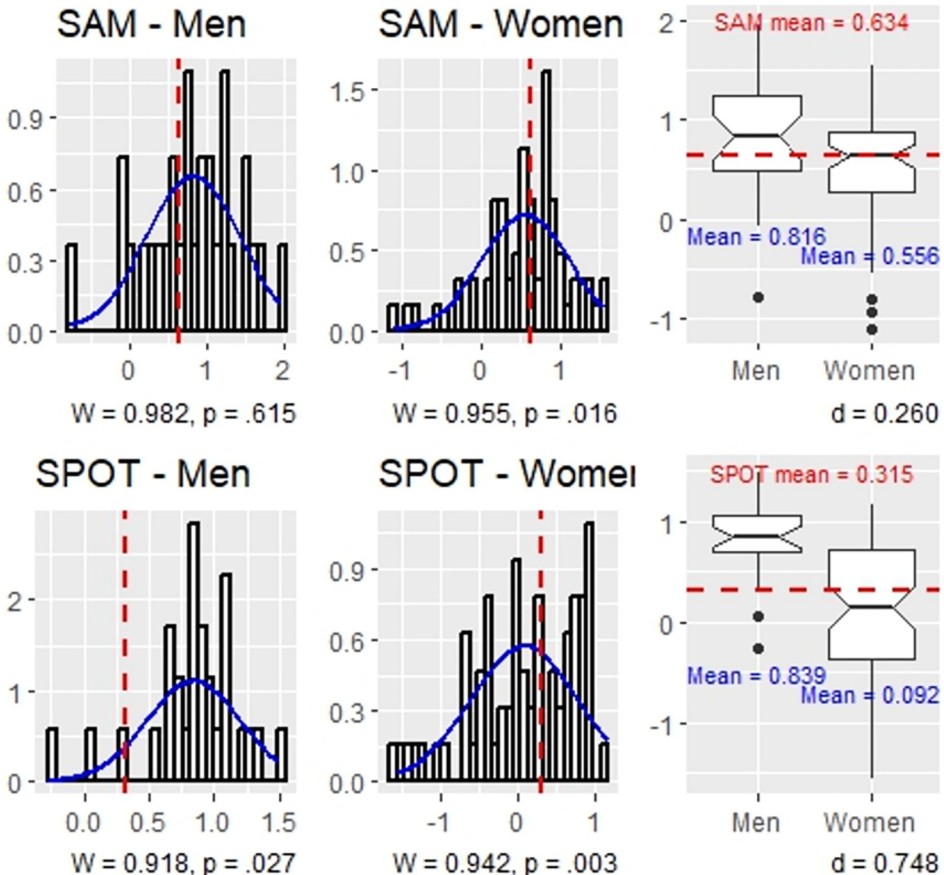

**Fig 5. The distribution of SAM and SPOT scores in men and women.**

not hypothesised) effects on the psychometrically measured constructs (SPOT, SAM, and CTt).

The parameters of the research model are presented in Table 4, while Table 5 summarizes the indirect effects of gender on computational thinking, as well as the contrasts of these effects. Fig 6 depicts the conceptual diagram of the research model.

**Table 4. Regression parameters and summary of the research model.**

| Antecedent | | Consequent | | | | | | | | | | |
|---|---|---|---|---|---|---|---|---|---|---|---|---|
| | | $M_1$ (SPOT) | | | | $M_2$ (SAM) | | | | Y (CTt) | | |
| | | b | SE | p | | b | SE | p | | b | SE | p |
| X (G) | $a_1$ | 0.749 | 0.139 | .001 | $a_2$ | -0.121 | 0.123 | .328 | c' | 2.595 | 0.908 | .005 |
| $M_1$ (SPOT) | | --- | --- | --- | $d_{21}$ | 0.521 | 0.080 | .001 | $b_1$ | 0.986 | 0.709 | .168 |
| $M_2$ (SAM) | | --- | --- | --- | | --- | --- | --- | $b_2$ | 3.327 | 0.763 | .001 |
| C (AGE) | | -0.002 | 0.024 | .949 | | -0.013 | 0.019 | .498 | | 0.093 | 0.138 | .503 |
| Constant | $i_{M1}$ | 0.126 | 0.528 | .813 | $i_{M2}$ | 0.783 | 0.406 | .058 | $i_Y$ | 11.172 | 3.064 | .001 |
| | | $R^2 = .240$ | | | | $R^2 = .346$ | | | | $R^2 = .433$ | | |
| | | $F(2, 94) = 14.739, p < .001$ | | | | $F(3, 93) = 16.404, p < .001$ | | | | $F(4, 92) = 17.558, p < .001$ | | |

*Notes.* Gender (X) coding 0 –women, 1 –men (positive coefficients indicate higher average values for men in paths involving X).

**Table 5. Tests of indirect effects of gender on CTt.**

|  | Path | Effect | SE (boot) | LLCI | ULCI | stat. sig. |
|---|---|---|---|---|---|---|
| Ind1 | G→SPOT→CTt | 0.738 | 0.481 | -0.159 | 1.720 | NO |
| Ind2 | G→SAM→CTt | -0.402 | 0.4137 | -1.237 | 0.413 | NO |
| Ind3 | G→SPOT→SAM→CTt | 1.298 | 0.393 | 0.6325 | 2.180 | YES |
| TOTAL | (Sum of indirect effects) | 1.634 | 0.573 | 0.540 | 2.795 | YES |
|  | Contrast | Effect | SE (boot) | LLCI | ULCI | stat. sig. |
| C1 | Ind1 minus Ind2 | 1.140 | 0.659 | -0.229 | 2.356 | NO |
| C2 | Ind1 minus Ind3 | -0.560 | 0.731 | -2.105 | 0.755 | NO |
| C3 | Ind2 minus Ind3 | -1.700 | 0.613 | -3.026 | -0.639 | YES |

*Notes.* Lower (LLCI) and upper (ULCI) limits of percentile confidence intervals are based on $N = 5000$ bootstrap samples. G: Gender; SPOT: Visuo-Spatial Ability; SAM: Fluid intelligence (Scrambled Adaptive Matrices); CTt: Computational Thinking test performance. Direct effect of gender on CTt = 2.595, $t(92) = 2.858$, $p = .005$. Total (direct plus indirect) effect of gender on CTt = 4.229, $t(92) = 4.514$, $p < .001$.

*H1*: *men perform better in computational thinking then women*. Hypothesis 1 is supported. Considering the simple association between gender and CT, men ($M = 19.379$, $SD = 4.747$) performed statistically significantly higher on the computational thinking test then women ($M = 15.118$, $SD = 3.908$) with a large effect size, $t(44.956) = 4.259$, $p < .001$, $d = 1.03$ ($U = 481$, $p < .001$, $r = .512$). On average, men completed 4 more CTt items correctly then women. Considering the path model, gender exerted its indirect effect on CT via visuospatial ability and fluid intelligence, accounting for 1.6 points in CTt performance (Table 5). However, gender's direct effect remained statistically significant, accounting for 2.6 points in CTt (Table 4, path *c'*; positive values indicate higher values for men). Based on the *t* statistic of the direct effect of G on CTt ($t = 2.951$) and the error degrees of freedom ($df = 95$), we calculated the effect size *r* for gender's effect on CT performance when accounting for gender differences in SAM and SPOT. The direct path (*c'*) had a small effect, $r = .176$. H1 is therefore supported: men performed statistically significantly better on the computational thinking test with small effect size when accounting for individual differences in visuospatial ability and fluid intelligence.

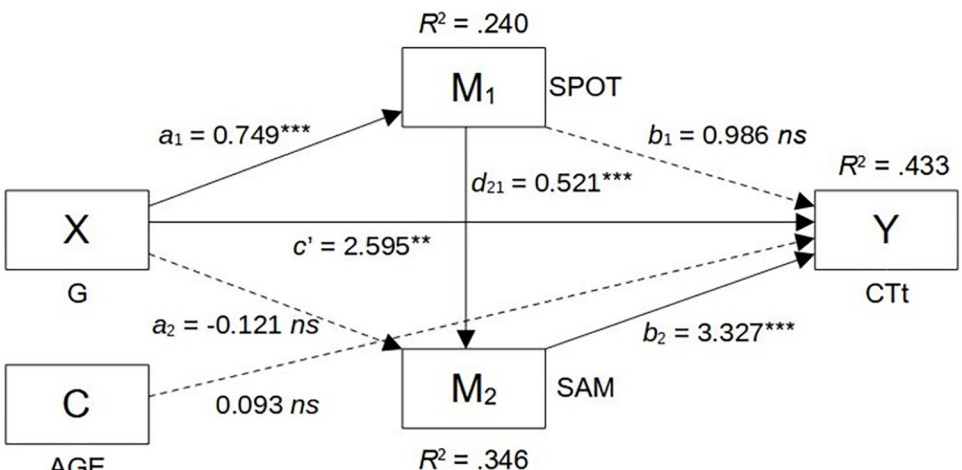

**Fig 6. Conceptual diagram of the research model.** *Notes.* **: $p < .01$; ***: $p < .001$; ns: Not statistically significant. SPOT: Visuo-Spatial Ability; SAM: Fluid intelligence (Scrambled Adaptive Matrices); CTt: Computational Thinking test performance. Paths from the covariant (C) to the mediators are omitted to promote readability (each path not stat. sig.).

*H2*: *men perform better in visuospatial ability then women*. Hypothesis 2 is supported. On average, men ($M$ = 0.839, $SD$ = 0.360) achieved statistically significantly higher standardized performance scores on the adaptive visuospatial ability test then women ($M$ = 0.092, $SD$ = 0.698) with a medium effect size, $t(91.417)$ = 6.931, $p <$ .001, $d$ = 0.748 ($U$ = 331, $p <$ .001). Reversing the mediation order between visuospatial ability and fluid intelligence in the path model showed that the effect of gender on visuospatial ability was statistically significant when controlling for fluid intelligence (and age), $b$ = 0.586, $t(93)$ = 4.961, $p <$ .001, $r$ = 0.457 (medium), lending further support to H2.

*H3*: *visuospatial ability is positively related to computational thinking ability*. Hypothesis 3 is supported. We found a large, statistically significant positive correlation between performance on the computational thinking test and visuospatial ability, $r$ = .53, $p <$ .001. The effect of visuospatial ability on computational thinking was mediated by fluid intelligence. Although visuospatial ability had a strong simple association with computational thinking performance, its direct effect was not statistically significant when accounting for the effect of fluid intelligence (Table 4, path $b_1$).

*H4*: *fluid intelligence is positively related to computational thinking ability*. Hypothesis 4 is supported. We found a large, statistically significant positive correlation between performance on SAM and CTt, $r$ = .56, $p <$ .001. With the effects of gender and visuospatial ability (and age) held constant, one standard deviation increase in fluid intelligence is associated with 3.3 points increase in CT performance (Table 4, path $b_2$).

Although we did not hypothesize gender differences in fluid intelligence, we note the average standardized performance score on the scrambled adaptive matrices test for men ($M$ = 0.816, $SD$ = 0.614) was 0.260 units higher than that of women ($M$ = 0.556, $SD$ = 0.558); this represents a small effect that nearly approached statistical significance, $t(48.693)$ = 1.9613, $p$ = .056, $ns$ ($U$ = 737, $p$ = .050). Path analysis showed that gender and fluid intelligence were spuriously associated: the simple association between gender and SPOT (see Tables 2 and 3) was accounted for by gender differences in visuospatial ability, $b$ = -0.121, $t(93)$ = -0.984, $p$ = .328, $ns$ (see Table 4, path $a_2$).

*H5*: *crystallized intelligence is positively related to computational thinking*. Hypothesis 5 is not supported. There was a small, statistically non-significant correlation between NoVo and CTt, $r$ = .13, $p$ = .206, $ns$ (see Table 3). Although NoVo moderately correlated with SPOT ($r$ = .28, $p$ = .005), it was uncorrelated with gender ($r$ = .10, $p$ = .325, $ns$) and SAM ($r$ = .12, $p$ = .233, $ns$).

*H6*: *the effect of gender on computational thinking performance is mediated by visuospatial ability*. Hypothesis 6 is not supported. Despite notable gender differences in visuospatial ability (see H2), the indirect effect of gender on CTt via SPOT was not statistically significant (Table 5, Ind1). Although gender's effect on CTt via SAM was also not statistically significant (Table 5, Ind2), its indirect effect through both mediators was statistically significant (Table 5, Ind3).

To compare the relative effects of fluid intelligence and visuospatial ability on CTt while controlling the effects of gender and age, we ran the analysis with standardized AGE and CTt variables to obtain the standardized effects of SPOT and SAM (note that SPOT and SAM scores are already in standard units), the partially standardized direct effect of gender, as well as the semipartial correlations squared to estimate the percent of variance in CT uniquely explained by each predictor. Fluid intelligence had the largest effect ($\beta$ = 0.72, $sr^2$ = .117), followed by gender ($\beta$ = 0.57 $sr^2$ = .050), SPOT ($\beta$ = 0.21, $ns$, $sr^2$ = .012) and AGE ($\beta$ = 0.05, $ns$, $sr^2$ = .003). Together, visuospatial ability, fluid intelligence, and gender differences accounted for over 43% of variance in computational thinking performance. Although controlled in the analyses, age had no effect (see Table 4, C (AGE) row).

## Discussion

### Summary of findings

Adults and students in higher education received relatively little attention in terms of their computational thinking performance and its relationship with underlying cognitive abilities. CT assessment is typically considered in the context of programming education, where the level of skill affecting CT performance makes it difficult to map and quantify the role of cognitive abilities, the influence of which may vary (presumably wane) according to building programming-related skills. While CT assessment in secondary education is especially important to inform course development and learning evaluation (see [12]), CT evaluation in adults is particularly relevant where we "[can] not rely on domain-specific knowledge and highly contextualized tasks" ([29], p. 1458). Apart from the utility of CT in professional education, the recent pandemic highlighted our increasing reliance on information systems to pursue everyday tasks and work assignments, even in adults who did not pursue specific interest in IT during their years in formal education. It is therefore timely to address CT measurement and its links to (cognitive) abilities that may support the development of computational thinking skills in adults.

We focused on visuospatial ability and fluid reasoning (an individual's ability that does not rely on previously acquired knowledge), as they have been shown to be related to CT performance. We used computerized adaptive tests developed on large samples based on item-response theory to measure these abilities, which allows for an efficient and precise assessment to yield standardized ability scores directly interpretable against a large population of test takers. We collected data in an individual and supervised setting to promote participants' focus on tasks. Data collection was organized in two sessions to help alleviate test fatigue and thereby derive test scores that closely reflect the level of the measured abilities. Variability due to level of programming skill was controlled by only considering individuals not familiar with programming languages. Beyond reporting simple associations between the demographic and psychometric constructs, we analyzed direct and indirect effects simultaneously in a path model, testing the effects of cognitive abilities on CT performance while controlling for the effects of gender and respondents' age. The present work represents the first use of the Computational Thinking test in the Hungarian language, as well as the use of computerized adaptive testing to measure fluid intelligence and visuospatial ability in the context of CT.

Research Question 1 was aimed at testing gender differences in CT in adults unfamiliar with programming. We found that men in our sample completed statistically significantly more items of the Computational Thinking test (CTt) correctly than women. Additionally, men's advantage in visuospatial tasks was confirmed.

Research Question 2 was aimed at exploring the relationship between CT performance, general intelligence, and visuospatial ability, controlling for the effects of gender and age. We found no statistically significant association between crystallized intelligence and CT. Fluid intelligence was a strong predictor of CT performance when controlling for the effects of gender, age, and visuospatial ability. Although men on average had higher visuospatial ability with large effect size, this gender difference did not directly influence CT performance, but it explained the small effect of gender on fluid intelligence. Visuospatial ability had a high correlation with CT performance, however, fluid intelligence explained its effect on CT, suggesting that visuospatial ability and CT are spuriously associated. This finding indicates that programming-naïve adults draw on their general fluid ability to solve computational thinking problems, which is not constrained specifically by visuospatial ability.

Since CTt items are fundamentally visuo-spatial in nature (problems presented in a maze or canvas format), Research Question 3 was aimed at testing whether the advantage of men in

CT performance can be accounted for by their widely reported general advantage in visuospatial ability (SPOT). The large univariate association between gender and CT was due to gender differences in visuospatial ability and fluid intelligence observed in our sample. Although men had higher visuospatial ability than women on average, the statistically non-significant indirect effect of gender on computational thinking through visuospatial ability indicates that gender differences in computational thinking are not accounted for by gender differences in visuospatial ability. Additionally, controlling for fluid intelligence rendered the direct effect of visuospatial ability on CT statistically non-significant. We note however that we measured fluid intelligence with the Scrambled Adaptive Matrices (SAM), which is a nonverbal test that relies on visuospatial presentation, just as the commonly applied Raven Progressive Matrices that was used to validate the test [59]. These findings suggest that non-programming adults were not constrained in their computational thinking performance due to their visuospatial abilities. When controlling for gender differences in visuospatial ability and fluid intelligence, the direct effect of gender on CT was small, which is in line with previous findings [14,20]. In particular, this result reproduces the findings on an adult sample who are CT novices and highlights the importance of considering general abilities in CT assessment in conjunction with gender effects in such abilities, especially when CT assessment relies on not gender-agnostic methods [28].

Additionally, gender had no statistically significant indirect effect on CT through fluid intelligence or visuospatial ability. However, the total indirect effect of gender through both cognitive factors was statistically significant; although lower in magnitude than the direct effect of gender on CT, the direct and indirect effects were not statistically significantly different (the confidence bounds of the total indirect effect contained the value of the direct effect; see Table 5). Since visuospatial ability and fluid reasoning did not fully explain the relative advantage of men in CT, further research is needed to elucidate this relationship. These findings are consistent with those of Guggemos [53], who proposed considering factors related to characteristics of students (e.g., computer literacy), their home environment, and their (formal) learning opportunities, as well as task characteristics, with other promising factors in explaining CT performance and outcomes including attitudes (e.g., see [21,27]) and personality [7].

Although fluid intelligence is a strong predictor of CT, the overlap between the two concepts is both empirically and theoretically partial. [12] points out that, similar to general intelligence, CT is broadly interpreted as an ability to solve complex problems. However, while intelligence is conceived as the result of the interaction of a wide array of cognitive processes [47], CT conceptually focuses on algorithmic operations in service of efficient and reusable solutions while utilizing the concepts of computer science [11]. CT can be interpreted in the context of the positive manifold, the all-positive pattern of correlations among performance on diverse cognitive tests [51], but with a focus of solving specific types of problems (algorithmic) that can be targeted with planned education programs to achieve favorable learning and performance outcomes.

Albeit fluid intelligence was a strong predictor of CT, their relationship was partial. The simple association between fluid intelligence and CT indicated a 31% overlap in variance ($r = .56$), while controlling for gender, visuospatial ability, and age decreased the percent of variance uniquely explained in CT by fluid intelligence to 12% ($sr^2 = .117$). Our participants were programming-naïve and could not rely on programming-specific knowledge to support their CT performance, therefore the above associations do not support conceptual or empirical near-equivalence of the two constructs. Additionally, our findings demonstrate an empirical separation between fluid intelligence and CT in terms of their structural relationships with other factors: gender differences in visuospatial ability accounted for gender differences in fluid intelligence, but not in CT. Thus, we argue that CT is not merely 'old wine in a new bottle', but its conceptual framework needs to incorporate (fluid) intelligence and other, more

specific measures of cognitive ability, in conjunction with level of programming skill, demographic factors, characteristics of CT tasks and measurement, non-cognitive factors (such as attitudes and personality), and possible interactions between them.

## Limitations and future work

Although the majority of CT research focuses on K-12, there is a growing need to consider adult populations to inform the development of professional education curricula [19,29], where testing novices informs the evaluation of education interventions and teacher training [52], and CT research in ascertaining a baseline level of relationship between CT and its underlying abilities. The present study controlled the level of programming-specific expertise by including only non-programmers as participants. We used convenience sampling, thus a distribution of abilities reflecting the studied population could not be achieved. However, we controlled demographic factors in our modelling approach, and the study was aimed at delivering a process-based argument focusing on the relationship between cognitive constructs rather than a population-based one [54]. Future research focusing on population-based inference should aim at a larger sample size. Despite the limited sample size and unbalanced gender distribution, our sample was appropriate for detecting medium effect sizes reported in the CT literature, and all statistically non-significant simple associations represented small effects. Future research could also consider conditioning the relationship between cognitive factors and CT on the level of programming skill, either in a categorical (non-programmer/programmer) or continuous manner (e.g., school grades). Although we controlled age as a covariate, the age range in our study was limited to young adults, thus age-dependent changes in underlying cognitive abilities (e.g., fluid intelligence; [68]), and their interactions with other factors (e.g., intelligence and gender; [69]) could not be tested. Future research aiming to model interactions and conditional processes should also consider expanding the measured cognitive and non-cognitive abilities in line with the growing CT literature to further clarify the relationship between intelligence, programming skills, and CT, while gender differences could be putatively explained by factors such as interest in programming [31], CT attitudes [21], and environmental and process factors [53] to expand the conceptual framework of CT.

## Conclusions

We measured computational thinking performance, visuospatial ability, and fluid intelligence in non-programming adults in higher education. Despite its large correlation with computational thinking (CT), visuospatial ability did not drive CT performance. Men had a small advantage over women in CT performance when holding the effects of cognitive abilities constant. Fluid intelligence was a strong predictor of computational thinking performance, suppressing the effect of visuospatial ability. We interpret this finding in the context of the Process Overlap Theory (POT), arguing that the nonverbal testing of fluid intelligence and performance on the Computational Thinking test (CTt) sample an overlapping set of underlying visuospatial processes. While non-programming adults rely on their general reasoning ability to solve CT problems, computational thinking conceptualizes their problem-solving skills supporting effective and efficient (algorithmic) solutions, an increasingly relevant set of skills amenable to improvement with targeted education programs.

## Supporting information

**S1 File.**
(CSV)

## Acknowledgments

We are grateful for the research assistants Petra Schrőter and Vivien Hartmann for their support in data collection.

## Author Contributions

**Conceptualization:** Gabor Aranyi.

**Data curation:** Gabor Aranyi.

**Formal analysis:** Gabor Aranyi.

**Funding acquisition:** Gabor Aranyi, Ferenc Kemény.

**Investigation:** Orsolya Pachner, Eszter P. Remete.

**Methodology:** Gabor Aranyi, Kristof Kovacs, Ferenc Kemény.

**Project administration:** Gabor Aranyi.

**Resources:** Kristof Kovacs, Balázs Klein.

**Visualization:** Gabor Aranyi.

**Writing – original draft:** Gabor Aranyi.

**Writing – review & editing:** Kristof Kovacs, Ferenc Kemény, Orsolya Pachner, Eszter P. Remete.

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
