## [Decision Letter · Decision Letter 0]

22 May 2024

PONE-D-24-03519Computational Thinking in University Students: The Role of Fluid Intelligence and Visuospatial AbilityPLOS ONE

Dear Dr. Aranyi,

Thank you for submitting your manuscript to PLOS ONE. After careful consideration, we feel that it has merit but does not fully meet PLOS ONE’s publication criteria as it currently stands. Therefore, we invite you to submit a revised version of the manuscript that addresses the points raised during the review process.

We look forward to receiving your revised manuscript.

Kind regards,

Liliana G Ciobanu

Academic Editor

PLOS ONE

Journal Requirements:

Gabor Aranyi received funding by the National Research, Development and Innovation Office of Hungary (NKFIH): Grant OTKA-FK-143095 project entitled ‘Advancing the Concept of Computational Thinking’. Kristof Kovacs received funding by the National Research, Development and Innovation Office of Hungary: Grant FK-21-138971, by the János Bolyai Research Scholarship of the Hungarian Academy of Sciences and by the ÚNKP-23-5 New National Excellence Program of the Ministry for Innovation and Technology from the source of the National Research, Development and Innovation Fund.

Balázs Klein is the executive director of Testar Ltd. The paper reflects the views of the scientists and not the company. The remaining authors declare that the research was conducted in the absence of any commercial or financial relationships that could be construed as a potential conflict of interest. 

We note that one or more of the authors are employed by a commercial company: Testar Ltd.

“The funder provided support in the form of salaries for authors, but did not have any additional role in the study design, data collection and analysis, decision to publish, or preparation of the manuscript. The specific roles of these authors are articulated in the ‘author contributions’ section.”

4. In the online submission form, you indicated that The data underlying the results presented in the study are available on request from the corresponding author (aranyi.gabor@ppk.elte.hu).

Reviewers' comments:

Reviewer's Responses to Questions

**Comments to the Author**

1. Is the manuscript technically sound, and do the data support the conclusions?

Reviewer #1: Partly

2. Has the statistical analysis been performed appropriately and rigorously? 

Reviewer #1: I Don't Know

3. Have the authors made all data underlying the findings in their manuscript fully available?

Reviewer #1: Yes

4. Is the manuscript presented in an intelligible fashion and written in standard English?

Reviewer #1: Yes

5. Review Comments to the Author

**Reviewer #1:** Computational Thinking in University Students: The Role of Fluid Intelligence and Visuospatial Ability

The study examines gender differences in computational thinking and how they are mediated by gender differences in visuo-spatial skills.

While I’ve enjoyed reading this paper, I had problems in understanding which was the specific focus of the study. The title and part of the introduction suggests that the focus is on the role of fluid intelligence and visuo.spatial skills, but when the reader comes to the HPs section and the analyses, the focus becomes unclear: Is it about the cognitive factors explaining performance in CT among university students unfamiliar with programming? In this case, gender should be a control/subject variable. Is the paper about the existence of gender-related differences in computational thinking and their relation with gender differences in visuo.spatial skills (as it seems from most of the analyses)? If so, this should be clarified from the beginning, and fluid and crystallized intelligence should just be control/covaried variables.

My general impression in reading the paper, and especially the HPs and analyses sections, was that the authors had problems in clarifying the specific role/nature of the study variables: which are the independent, dependent, mediation and control variables of the study?

Morevoer, not all the analyses follow up from the RQs of the study and the literature review. While the mediational role of visuo.spatial skills in CT is supported by prior literature, the reason why the authors are assuming a mediation/moderation role of fluid intelligence explaining gender differences in CT is less clear.

Additionally, some analyses seem to follow from revisited? HPs rather than the original HPs of the study: for instance, the analyses testing whether gender differences in fluid intelligence are accounted for by gender differences in visuospatial ability (SPOT) (p. 18)

I did not understand why did the authors consider revisited HPs or what they do mean with “revisited”? Are they ex-post HPs?

Sample. Although the authors recognize they have considered a convenience sample of psychology students, given that the prevalence of boys and girls enrolled in psychology classes is typically unbalanced, especially in some psychology classes, they could have involved more courses or classes of psychology to increase the number of boys in their sample (only 29 young men, versus 68 young women).

Procedure. For all standardized measures used, reliability/validity should be reported

6. PLOS authors have the option to publish the peer review history of their article (what does this mean?). If published, this will include your full peer review and any attached files.

Reviewer #1: No

---

## [Author Response · Author response to Decision Letter 0]

3 Aug 2024

Response to reviewers

We are grateful for Reviewer #1 for the useful comments. Addressing these comments has helped us to substantially improve the manuscript. The Reviewer’s comments are reproduced verbatim below with our answers inserted where appropriate.

---

Reviewer #1: Computational Thinking in University Students: The Role of Fluid Intelligence and Visuospatial Ability

The study examines gender differences in computational thinking and how they are mediated by gender differences in visuo-spatial skills.

While I’ve enjoyed reading this paper, I had problems in understanding which was the specific focus of the study. The title and part of the introduction suggests that the focus is on the role of fluid intelligence and visuo.spatial skills, but when the reader comes to the HPs section and the analyses, the focus becomes unclear: Is it about the cognitive factors explaining performance in CT among university students unfamiliar with programming? In this case, gender should be a control/subject variable. Is the paper about the existence of gender-related differences in computational thinking and their relation with gender differences in visuo.spatial skills (as it seems from most of the analyses)? If so, this should be clarified from the beginning, and fluid and crystallized intelligence should just be control/covaried variables.

With thanks to the reviewer, we clarified in the manuscript that the focus remains on the role of cognitive abilities, as in the title. However, since there is strong evidence in the literature for gender differences in both visuospatial ability and CT performance, gender is controlled in the analyses. We added an extra paragraph to the Background sub-section to emphasize the focus of the research:

The present research investigates the relationship between CT performance, general intelligence, and visuospatial ability in students in higher education. The following sections provide an overview of CT measurement and the relevant research addressing the role of visuospatial ability and general intelligence in driving CT performance. We argue that a better understanding of CT and its relationship with well-established cognitive factors is supported by controlling for the effects of potential confounds which we identify in previous literature, such as participants’ age, their programming experience, and widely-reported gender differences in visuospatial ability and in CT performance.

We made some further changes in the Introduction section, and the last paragraph of the updated Results section contains the relative contributions of these variables to CT performance, emphasizing the role of fluid intelligence. We also refocused the paper by explicitly stating the research question related to the relationships between CT and cognitive abilities:

RQ2: How is computational thinking related to visuospatial ability, fluid intelligence, and crystallized intelligence in adults unfamiliar with programming?

My general impression in reading the paper, and especially the HPs and analyses sections, was that the authors had problems in clarifying the specific role/nature of the study variables: which are the independent, dependent, mediation and control variables of the study?

We confirm that the focus of the paper is on the role of fluid intelligence and visuospatial ability on CT. However, due to gender differences in both visuospatial ability and CT widely reported in the literature, as well as to correlations expected between the cognitive constructs (visuospatial ability, fluid IQ, and crystallized IQ) discussed in the positive-manifold and process-overlap arguments, we argue in the manuscript that gender as a confounding factor needs to be controlled. With thanks to the reviewer, we have made changes in the manuscript to clarify the focus (see above comment). We have also made changes to the Research Questions and Hypotheses section to highlight the role of gender as control variable:

Gender is treated as a control variable in the present study, the effects of which on other study variables are considered when examining the relationship between CT and cognitive factors. Thus, H1 is formulated to test if gender differences in CT often reported in younger participants can be observed in programming-naïve university students, while H2 aims at testing gender differences in visuospatial ability often reported in the cognitive literature.

Morevoer, not all the analyses follow up from the RQs of the study and the literature review. While the mediational role of visuo.spatial skills in CT is supported by prior literature, the reason why the authors are assuming a mediation/moderation role of fluid intelligence explaining gender differences in CT is less clear.

With thanks to the reviewer, we note that no a priori hypothesis is related to the mediating effect of fluid intelligence between gender and CT. We made changes to the Research Questions and Hypotheses section related to the mediation hypothesis (H6) to clarify. We have also dropped RQ3 (the previous RQ2 is now RQ3, because a new RQ2 was added to explicitly address the relationships between CT and cognitive factors). Although a direct path from gender to CT is specified based on the simple association between gender and SAM, this is only used to test if the association between the two are spurious (see the updated H4 report and the comment below), which is supported by our results. Conversely, despite the large correlation between SAM and SPOT, fluid intelligence does not explain gender differences in visuospatial ability (see the updated H2 report).

Additionally, some analyses seem to follow from revisited? HPs rather than the original HPs of the study: for instance, the analyses testing whether gender differences in fluid intelligence are accounted for by gender differences in visuospatial ability (SPOT) (p. 18)

Indeed, this analysis association was not hypothesized, but we used the path model to follow up the simple association between gender and fluid intelligence reported in tables 2 and 3; considering visuospatial ability showed that gender and fluid intelligence were spuriously associated. We clarified this in the updated section related to H4.

I did not understand why did the authors consider revisited HPs or what they do mean with “revisited”? Are they ex-post HPs?

With thanks to the reviewer, we substantially reorganized the Results section of the manuscript. We now report hypothesis tests in one place, following the presentation of descriptive statistics, tables reporting simple associations, and the path model. Hypothesis tests are now reported in the Hypothesis tests and Indirect Effects sub-section, while the summary tables of simple associations are kept in the Descriptive Analysis and Simple Associations sub-section.

Sample. Although the authors recognize they have considered a convenience sample of psychology students, given that the prevalence of boys and girls enrolled in psychology classes is typically unbalanced, especially in some psychology classes, they could have involved more courses or classes of psychology to increase the number of boys in their sample (only 29 young men, versus 68 young women).

We agree with the reviewer that larger samples are desirable in general. As our protocol concentrated on controlled data collection in an individual and supervised setting consisting of two sessions, we aimed at achieving the required sample size to test a priori hypotheses informed by the effect sizes reported in the reviewed literature. In other words, we aimed to maximize data quality over quantity. We discuss in the Limitations and Future Work section that future studies aiming at population-based inferences should recruit a larger sample. We made additions to the Participants and Limitations and Future Work sections to expand on this.

Procedure. For all standardized measures used, reliability/validity should be reported

Thank you for pointing out that no information was provided about the tests’ accuracy. However, since the adaptive tests rely on item response theory (IRT), the classical notion of test reliability cannot be applied. This is because unlike in classical tests where accuracy is a test property represented in the reliability statistic, under IRT the error margin (standard error, SE) of each theta value is estimated. We added the mean SE for each adaptive test to Table 1. Additionally, since classical reliability values can be approximated from IRT-based SE values, we have calculated these values and include them in the main text:

The computerized adaptive tests are based on item response theory (IRT), where the conception of accuracy is different from the concept of reliability in classical test theory. That is, under the IRT framework it is the (item or test) information function that represents precision so that information is the inverse of the items’ or the test’s accuracy: the more information the test or the items provide, the smaller the error margin of the measurement. Under IRT models the SE (standard error) of each ability (theta) score can be calculated; the higher the information the smaller the SE. We report the mean SEs obtained for each computerized adaptive test in Table 1. Classical reliability coefficients can be approximated from SE values obtained in IRT [66, 67]. The reliability values estimated from the SE values that appear in Table 1 are .92, .95, and .97 for SPOT, SAM, and NoVo, respectively.

References to papers that report the psychometric properties and the validity of SAM and NoVo have been added to the Materials and Methods section (321-2., 331-2.). Unfortunately, validity information for the SPOT test is not yet available.

For the validity of the computational thinking test, we refer to [14]. Unfortunately, validity information for the Hungarian version of CTt is not yet available.

Additionally, a detailed check of our manuscript revealed that the constants in Table 4 were erroneously recorded (all other parameters were correctly displayed). Therefore, we have replaced Table 4 in the updated manuscript; the table now contains the correct values. We have also changed a misleading abbreviation in Table 5 (we now use the test’s name, SPOT, instead of VSA for visuospatial ability).

---

## [Editor Report · Decision Letter 1]

13 Aug 2024

Computational Thinking in University Students: The Role of Fluid Intelligence and Visuospatial Ability

PONE-D-24-03519R1

Dear Dr. Aranyi,

We’re pleased to inform you that your manuscript has been judged scientifically suitable for publication and will be formally accepted for publication once it meets all outstanding technical requirements.

Kind regards,

Liliana G Ciobanu

Academic Editor

PLOS ONE
---

## [Editor Report · Acceptance letter]

23 Aug 2024

PONE-D-24-03519R1 

PLOS ONE

Dear Dr. Aranyi, 

I'm pleased to inform you that your manuscript has been deemed suitable for publication in PLOS ONE. Congratulations! Your manuscript is now being handed over to our production team.

Kind regards, 

on behalf of

Dr. Liliana G Ciobanu 

Academic Editor

PLOS ONE